# Characterization of Oral *Veillonella* Species in Dental Biofilms in Healthy and Stunted Groups of Children Aged 6–7 Years in East Nusa Tenggara

**DOI:** 10.3390/ijerph192113998

**Published:** 2022-10-27

**Authors:** Citra Fragrantia Theodorea, Saint Diven, Devin Hendrawan, Ariadna Adisattya Djais, Boy Muchlis Bachtiar, Armelia Sari Widyarman, Chaminda Jayampath Seneviratne

**Affiliations:** 1Department of Oral Biology, Faculty of Dentistry, Universitas Indonesia, Jalan Salemba Raya No. 4, Central Jakarta 10430, Indonesia; 2Undergraduate Program, Faculty of Dentistry, Universitas Indonesia, Jalan Salemba Raya No. 4, Jakarta 10430, Indonesia; 3Department of Microbiology, Faculty of Dentistry, Trisakti University, Jalan Kyai Tapa No. 1, West Jakarta 11440, Indonesia; 4National Dental Research Institute Singapore, National Dental Centre Singapore, Singapore 168938, Singapore; 5Oral Health Academic Clinical Programme, Duke-NUS Graduate Medical School, Singapore 169857, Singapore

**Keywords:** oral *Veillonella*, stunting, oral hygiene status, dental biofilm, East Nusa Tenggara

## Abstract

Impaired development that causes stunting is one of the most common health problems in Indonesia. In particular, the highest number of cases of stunting in Indonesia was reported in the East Nusa Tenggara (NTT) province. Previous studies have shown a tendency for deteriorating oral hygiene in children with a poor nutritional status. In addition, a higher proportion of oral *Veillonella* has been reported in children with poor oral hygiene. However, the relationship between populations of oral *Veillonella* and stunting has not been studied before. Therefore, this study aimed to analyze the oral *Veillonella* profile in the dental biofilms of healthy and stunted children aged 6–7 years. The participants were 60 elementary school students in the Nangapanda District, Ende, NTT, Indonesia. In this study, real-time polymerase chain reaction was used to examine dental biofilm samples from the healthy (n = 31) and stunted (n = 29) groups. The results revealed that seven oral *Veillonella* species were found in all groups. However, the number of four oral *Veillonella* species significantly differed between the healthy and stunted groups: *V. denticariosi*, *V. infantium*, *V. rogosae*, and *V. tobetsuensis*. This is the first study to demonstrate a potential association between oral *Veillonella* species and stunting in children.

## 1. Introduction

The oral cavity has various unique and complex components, such as teeth, the tongue, mucous membranes, and fluids that are physiologically produced by the body, which are also called the oral environment [1]. When healthy, the oral environment is characterized by a temperature ranging from 33.2 to 38.2 °C, a relatively neutral pH level, well-maintained composition and continuity of salivary flow, and balanced oral bacterial composition [1,2,3]. However, unbalanced conditions, such as changes in temperature, pH level, and salivary flow, may increase the number of certain bacteria and lead to multispecies communities on tooth surfaces, which are known as dental biofilms [4,5].

Dental biofilms can cause various diseases in the oral cavity, including caries and periodontitis [4]. Dental biofilm formation occurs chronologically, starting from the initial, early, middle, and late colonization stages [6,7]. Oral *Veillonella* species are found in 10% of dental biofilms during the early colonization stage [8]. At this stage, oral *Veillonella* species colonize with *Streptococcus* species to form dental biofilms [9]. Bacteria belonging to the oral *Veillonella* species group are *Veillonella atypica*, *Veillonella denticariosi*, *Veillonella dispar*, *Veillonella parvula*, *Veillonella rogosae*, *Veillonella tobetsuensis*, and *Veillonella infantium* [10,11].

A previous study has stated that the number of oral *Veillonella* species significantly increased along with poor levels of oral hygiene [10]. Pustelny et al. showed that the presence of *V. parvula* could lead to a higher resistance of cocci bacteria, such as *Streptococcus mutans*, counter chemical antiseptics [12]. In addition, deteriorating oral hygiene status in children may affect food selection and nutrient intake, leading to malnutrition [13]. Many studies have demonstrated how the oral–gut microbiome axis can regulate the pathogenesis [14]. In relation to *Veillonella* species, Zhan et al. reported that an overabundance of *Veillonella* promotes intestinal inflammation [15]. It is evident that *Veillonella* species might have a role in pathogenesis, yet the mechanism has not been fully clarified.

Some studies have found that poor nutrition is influenced by the high number and frequency of consumption of foods containing sugar [16]; a lack of consumption of vitamins A, C, E, and folic acid [17]; and body composition [18]. However, indicators used to determine the nutritional status of individuals remain unclear.

Body composition measurements were qualitatively performed using anthropometric measurements. The results of anthropometry measurements can show whether a person has weight and height appropriate to age or vice versa [18,19]. This indicator has the advantage that it can comprehensively measure all aspects of growth influenced by nutritional status; hence, the measurement value is valid and in accordance with the protocol standards used globally [20]. If someone has a height that is clinically and quantitatively short compared to their age, they can be categorized as groups with a stunting status [21]. Stunting, among others, can cause a decline in cognitive abilities, memory, and locomotor skills and increase the risk of mortality [21].

In Indonesia, the highest number of stunting cases was found in the province of East Nusa Tenggara (NTT), with a prevalence of 41.1% [22]. The Indonesian government has reduced the prevalence of stunting in children through the National Strategy for the Acceleration of Stunting Prevention [23,24]. One of the efforts to support the program was to increase research on stunting preventive measures. From the aspect of dentistry science, researchers see that there is a phenomenon of increase in the number of oral *Veillonella* in children at the level of oral hygiene. In addition, poor oral hygiene is associated with poor nutritional status [16]. However, the relationship between the number of oral *Veillonella* species and stunting status is not yet known. These results may be useful clinical bioindicators of nutritional status. Based on this description, this research was conducted to analyze the oral *Veillonella* profile of dental biofilms in healthy and stunted children aged 6–7 years.

## 2. Materials and Methods

### 2.1. Ethical Statement

This study was approved by the Health Research Ethics Committee of the Faculty of Medicine, Universitas Indonesia, Cipto Mangunkusumo Hospital, Jakarta, Indonesia. The approval number was KET-1060/UN2.F1/ETIK/PPM.00.02/2019.

### 2.2. Participants

This was a cross-sectional descriptive study. A convenience sample of 60 elementary school children, aged 6–7 years old, from Nangapanda District, Ende, NTT, Indonesia, were voluntarily recruited in October 2019. The medical history of the children was obtained from their parents. Children with mental retardation, cleft lip and palate abnormalities, and those on antibiotics and antihistamines in the last three months were excluded from this study. All participants and their parents received an explanation regarding the aim and procedures of this study for their agreement to participate. Written informed consent was obtained from the parents of all participants. Five dentists received a 4 h training session, during which they were instructed by a senior faculty member, who served as the gold standard to avoid potential bias. After training, all five dentists screened the 10 trial patients. The agreement among all examiners regarding the Simplified Oral Hygiene Index and body composition was 100% (kappa statistic = 1.0). Sample sites were dried carefully and were isolated from saliva contamination by cotton rolls. The supragingival dental plaque sample was collected from the first permanent molar surface using a dry sterile cotton bud and dissolved into 1 mL of sterile Phosphate-Buffered Saline (PBS). Samples were stored at −4 °C for no longer than 6 h and then moved to −80 °C storage.

### 2.3. Nutritional Status

Weight was measured to the nearest 0.1 kg using a SECA digital weighing scale. Standing height was measured to the nearest 0.1 cm using a microtoise. HAZ was used to determine stunting (height-for-age Z-score < −2.00). Stunting was defined as a height-for-age z-score based on the WHO Child Growth Standard [25].

### 2.4. Simplified Oral Hygiene Index

The study participants were also divided into three groups based on their evaluation by the Simplified Oral Hygiene Index (OHI-S): good, moderate, and poor hygiene groups, according to the criteria of Greene and Vermilion [26].

### 2.5. DNA Extraction

Genomic DNA was extracted from the isolated bacterial cells using an Insta Gene Matrix Kit (Bio-Rad Laboratories, Hercules, CA, USA). The DNA concentration was determined based on fluorescence using a Qubit 3.0 fluorometer (Invitrogen, Carlsbad, CA, USA), according to the manufacturer’s protocol.

### 2.6. Identification of Oral Veillonella spp.

Specific primer sets of oral *Veillonella* species were used in this study for the identification of Veillonella species [27,28].

### 2.7. Real Time-qPCR Protocol (RT-qPCR)

First, the master mix was provided with 6 μm (0.75 μL/well) forward primers, 6 μm reverse primers (0.75 μL/well), SYBR Green PCR Mix (0.75 μL/well), and Nuclease-free water (1 μL/well). Furthermore, 5 μL of the sample (duplicates), and 10 μL of the master mix was added to each well. Another duplicate contained 5 μL NFW, and 10 μL master mix was introduced as a non-template control (NTC).

The RT-qPCR engine was run in the following sequence of steps: 95 °C for 2 min (1×), 95 °C for 5 s, and 60 °C for 20 s (40×).

The number of bacteria was obtained by entering the cycle threshold (CT Mean) value of the RT-qPCR reading of each sample into the formula of the standard curve (Appendix A). The formula for each standard curve is as follows [29]:

CT-Mean Convert to Log
*Y = mx + b*, → x = $\frac{\left(Y-b\right)}{m}$Y→ CT Mean each samplem→ slopeb→ intercept**Log (number of bacteria) = 10^x^**


### 2.8. Data Analysis

Comparative and correlation analyses of the number of oral *Veillonella* species in dental biofilm samples obtained from the subjects was performed using RT-qPCR. The data obtained was then tested using Mann–Whitney and Spearman statistics with the IBM^®^ SPSS program (version 23.0) (IBM Corp., Armonk, NY, USA).

## 3. Results

Based on nutritional status, the number of children with normal HAZ was 51.7%, whereas that of children with stunting was 48.3%. The prevalence of children with normal and stunting status seems to be quite balanced. Based on the oral hygiene status, 16.7% of children had good oral hygiene, 45% had moderate oral hygiene, and 38.3% had poor oral hygiene. Demographic characteristics of the participants are presented in Table 1.

In a total of 60 children with normal and stunting status, it was found that children in the normal group have 19.4% good OHI-S, 45.2% moderate OHI-S, and 35.5% poor OHI-S. Meanwhile, 13.8% of children in the stunted group had good OHI-S, 44.8% had moderate OHI-S, and 41.4% had poor oral hygiene (see Figure 1).

All oral *Veillonella* species were detected in dental biofilm samples. *V. atypica*, *V. denticariosi*, *V. infantium*, *V. rogosae*, and *V. tobetsuensis* were abundant, whereas *V. dispar* and *V. parvula* were found in low numbers among others (Figure 2).

Furthermore, we compared oral *Veillonella* species based on the nutritional status. All oral *Veillonella* species were detected in dental biofilm samples. However, *V. parvula* was found to have the lowest abundance. In addition, the seven oral *Veillonella* species showed that the data were not normally distributed; hence, non-parametric statistical tests were used. According to the comparative Mann–Whitney statistical test, the number of four oral *Veillonella* species between the stunted and normal groups differed significantly. *V. denticariosi*, *V. infantium*, and *V. rogosae* were less abundant in the stunted group than in the normal group (*p* = 0.00, *p* = 0.00, and *p* = 0.02, respectively). Meanwhile, *V. tobetsuensis* was found to be more abundant in the stunted group than in the normal group (*p* = 0.04) (see Figure 3).

Comparison of the number of seven species of oral *Veillonella* based on OHI-S revealed a significant difference in *V. denticariosi*, which was found to be lower in moderate and poor oral hygiene than in good oral hygiene (*p* < 0.05). *V. infantium*, *V. rogosae*, and *V. tobetsuensis* were progressively higher in deteriorating oral hygiene status (*p* < 0.05) (Figure 4).

## 4. Discussion

One of the health problems faced by children in Indonesia is stunting, a growth disorder. The condition of stunting inhibits development, resulting in decreased intellectual and physical abilities and even increased symptoms of depression. The main cause of stunting is a lack of nutritional supply during the first 0–1000 days of life. According to several previous studies, poor OHI-S scores have an impact on stunting and underweight. Children with poor OHI-S scores experience discomfort or even pain when eating, resulting in disturbances in eating patterns and nutritional intake. Poor OHI-S is generally caused by inadequate knowledge of parents on oral health maintenance, which is exacerbated by their low socioeconomic status. Children who grow up in families with low socioeconomic status have poor OHI-S, which affects the value of oral impact on daily performance (OIDP) [30]. The low intake of nutrients from outside the body has an impact on the growth and development of children and directly affects systemic conditions, among which is the changes in the gastrointestinal (GI) tract, including the oral environment. Bacteria in the oral environment, as a part of the GI tract, also undergo changes. Vonaesch et al. (2017) stated that children who lack nutritional intake and suffer from stunting experience has an increase of Gram-negative bacteria number (*Escherichia coli*, *Shigella* spp., and *Campylobacter* spp.) [31]. In the small intestine, there is an increase in the number of bacteria that usually reside in the oropharynx. In addition, Dinh et al. (2016) reported an increase in the number and variety of GI tract bacteria in children with stunting status [32].

This phenomenon led us to determine the number of oral bacteria in the GI tract of children with stunting conditions. One of the Gram-negative bacteria found in the oral cavity is oral *Veillonella* spp. According to Mashima et al. (2015), oral *Veillonella* can be found in dental biofilms during the early colonization stages [9]. In this study, the age range of the subjects was 6–7 years and the dental biofilm samples were taken from the surface of the mandibular first molar. Based on the chronological theory of permanent teeth growth, at that age, the permanent mandibular first molar just erupts and is exposed to the oral environment; hence, the dental biofilm formed is expected to be in the early colonization stage. To determine the amount of oral *Veillonella* in the dental biofilm, absolute quantification was performed using a specific primer set with the RT-qPCR machine, where this type of quantification was compared after converting the CT Mean of each sample to log10 CFU/mL.

In this study, 60 samples of dental biofilm for children were analyzed, with the proportion of gender: boys = 43.3% and girls = 56.7%, age: 6 years = 23.3% and 7 years = 76.7%, OHI-S: good = 16.7%, moderate = 45%, and poor = 38.3%, and normally nourished children = 51.7% and stunted children = 48.3%. In the normal and stunted groups, the number of children with moderate and poor OHI-S scores was the same, but there were fewer children with good OHI-S scores in the stunted group. From the whole sample, the bacteria that were dominant (higher) compared to other oral *Veillonella* species were *V. atypica*. This result is different from that of the previous studies conducted by Theodorea et al. (2017) and Djais et al. (2019) using the sample of saliva from children (which were also divided into three OHI-S categories), where the number of *V. atypica* did not appear as the most abundant species [10,33]. One of the factors that may have contributed to this distinction is the difference in the isolation of the niche microenvironment.

Based on the nutritional status, only *V. denticariosi*, *V. infantium*, *V. rogosae*, and *V. tobetsuensis* were significantly different. Children with healthy nutritional status had a higher number of *V. denticariosi, V. infantium*, and *V. rogosae* than the stunted group, while *V. tobetsuensis* was higher in the stunted group. Additionally, we compared the number of oral *Veillonella* species based on oral health status. The results showed significant differences between the oral numbers of *Veillonella* species at good, moderate, and poor OHI-S values, including *V. denticariosi*, *V. infantium*, *V. rogosae*, and *V. tobetsuensis.* This phenomenon revealed the potential of these four oral *Veillonella* species in both the oral environment and body.

According to the optimum oral environment for *Veillonella*, these bacteria live in association with lactic acid-producing species, such as *Streptococcus mutans*, during the early colonization stage of dental biofilms, which causes the continuation of colonization by dental biofilm bacteria and decreases the pH of the oral environment to become acidic. The high number of *S. mutans*, which is the main bacterium that causes caries, is an indicator of poor oral hygiene (measured by a person’s OHI-S score). Based on research conducted by Dimaisip et al. (2018), poor OHI-S scores have an impact on stunting and underweight conditions. However, in this study, the sample used was from NTT Province, which is not only the province with the highest prevalence of stunting but also with a very high prevalence of oral dental health problems (approximately 54.9%) and very poor knowledge of brushing teeth (only 3.7% of children brushed their teeth at the right time). This causes the average OHI-S value in normal and stunted children to not differ significantly; in fact, the distribution is fairly even. In addition, the level of cleanliness or sanitation of NTT is still low. Riskesdas 2018 data show that only 20.4% of people of NTT have knowledge of proper hand washing, which is less than the national average by 29.4%. These factors could be the cause of the differences in the results of research data with similar studies in different countries (such as Thailand and Japan).

In this study, there were several novelties, including the first time that the RT-qPCR machine was used to analyze oral *Veillonella* spp. with species-specific primer for *Veillonella* based on the *rpoB* and *dnaK* gene [11,27,28] from dental biofilm samples, and for the first time, an analysis of the relationship between the amount of oral *Veillonella*, stunting status, and oral hygiene index in Indonesia was conducted.

The limitations of this study include the limited data from previous research on specific oral *Veillonella* species in Indonesia. Additionally, it is possible to increase the number of samples to improve the validity of the results of this study. It is hoped that the results of this study will be useful for further research, particularly for *V. denticariosi*, *V. infantium*, *V. rogosae*, and *V. tobetsuensis,* which might have potential as clinical bio-indicators of stunting status. Because stunting is a chronic syndrome, a longitudinal study rather than a cross-sectional approach would allow us to better assess the causal relationship between oral bacteria and stunting in the future.

## 5. Conclusions

This study demonstrates the potential of *V. denticariosi*, *V. infantium*, *V. rogosae*, and *V. tobetsuensis* as clinical indicators of stunting status. However, future studies on this species are required.

## Figures and Tables

**Figure 1 ijerph-19-13998-f001:**
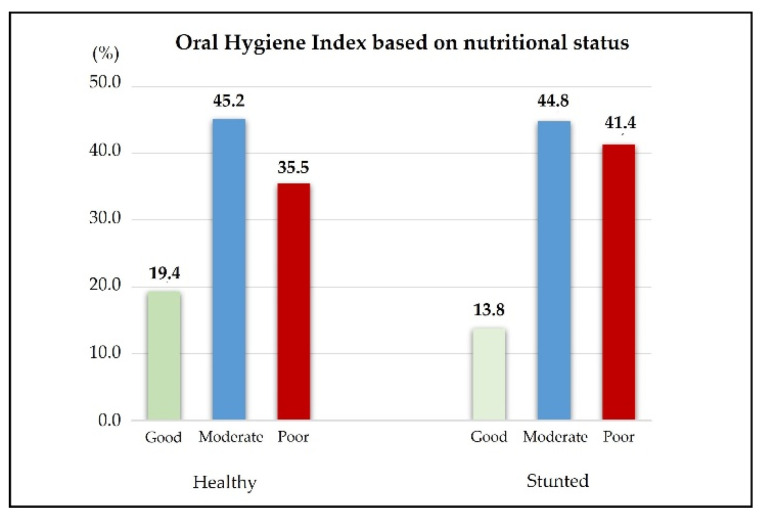
Percentage of oral hygiene group based on nutritional status.

**Figure 2 ijerph-19-13998-f002:**
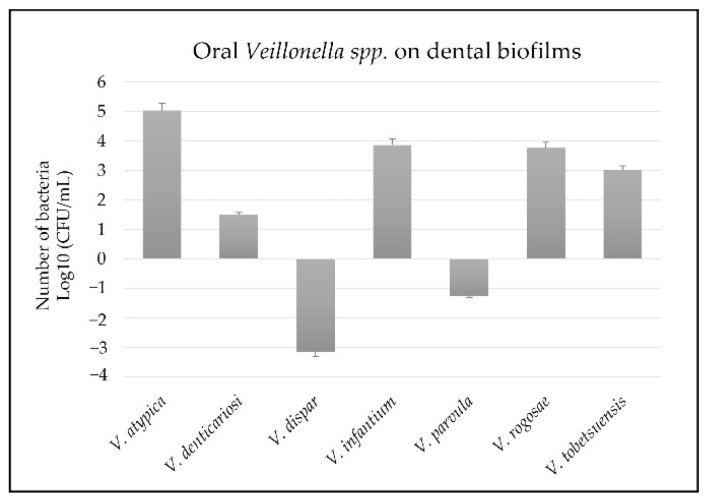
Number of oral *Veillonella* species on dental biofilms.

**Figure 3 ijerph-19-13998-f003:**
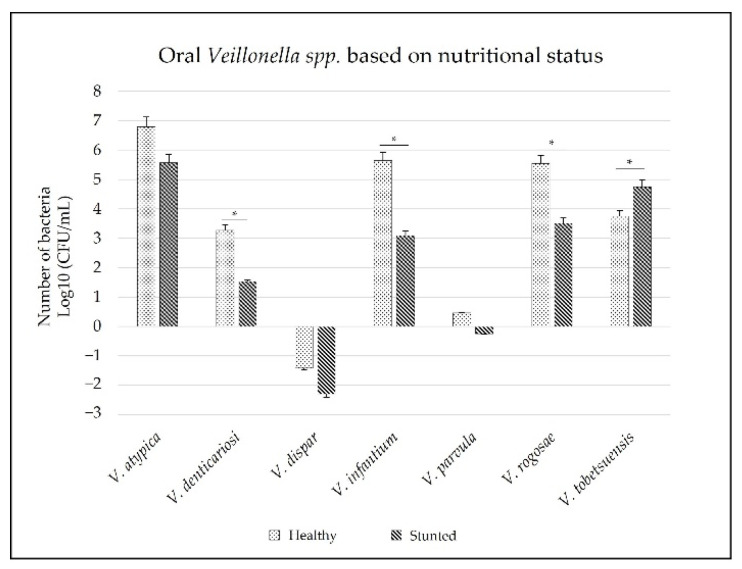
Number of oral *Veillonella* species based on the nutritional status. (*) Statistically significant (*p* = 0.05).

**Figure 4 ijerph-19-13998-f004:**
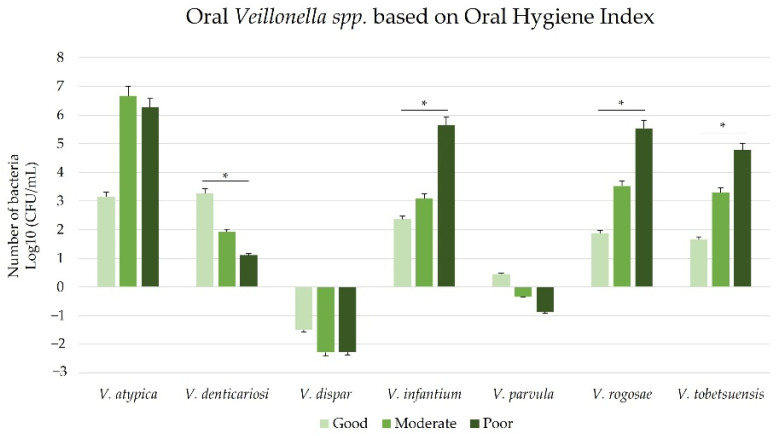
Number of oral *Veillonella* species based on oral hygiene status. (*) Statistically significant (*p* = 0.05).

**Table 1 ijerph-19-13998-t001:** Participants’ characterization.

Characteristics		Frequency	Percentage
		(*n*)	(%)
Gender	Boys	26	43.3%
	Girls	34	56.7%
Age	6 years old	14	23.3%
	7 years old	46	76.7%
Nutritional Status	Normal	31	51.7%
	Stunting	29	48.3%
Oral Hygiene	Good	10	16.7%
	Moderate	27	45%
	Poor	23	38.3%

## Data Availability

The datasets analyzed during the current study are available from the corresponding author upon request.

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
