# Peer review of "Characterization of Oral *Veillonella* Species in Dental Biofilms in Healthy and Stunted Groups of Children Aged 6–7 Years in East Nusa Tenggara"

_ijerph, 2022, doi:10.3390/ijerph192113998_

Round 1
Reviewer 1 Report
Why did the author decide to focus the work on veillonella? The justification for the work is missing, that is, the mouth has a very large biofilm, if you mention that only in 10% of the cases does veillonella occur, what is the importance of focusing on it, in addition to the fact that the pathogenic agent is not specific to the child population .
Author Response
Response to Reviewer 1 Comments
Point 1: Why did the author decide to focus the work on veillonella? The justification for the work is missing, that is, the mouth has a very large biofilm, if you mention that only in 10% of the cases does veillonella occur, what is the importance of focusing on it, in addition to the fact that the pathogenic agent is not specific to the child population.
Response 1: The importance of Veillonella spp. were found in some studies. It was report that the presence of V. parvula leads to a higher resistance of cocci, such as Streptococcus mutans, against chemical antiseptics. Recent studies have demonstrated that the oral-to-gut and gut-to-oral microbial transmission can regulate pathogenesis, indicating the presence of the oral–gut microbiome axis. Veillonella species were found over abundance in gastrointestinal inflammation. It is evident, that Veillonella species might have role in pathogenesis yet the mechanism have not been fully clarified. (it has been stated in paragraph 3)

Reviewer 2 Report
Dear authors,
thanks for having submitted such an interesting manuscript.
My main concern is about the null hypothesis. Reading from the introduction the authors aimed at finding a sort of relationship between stunting and Oral Veillonella colonization. Considering that malnutrition is a well known causative factor for stunting and that Veillonella spp. can be found (even if not exclusively) in oral cavities of people with bad oral hygiene what seems to be lacking is the eventual relation between bacterial colonization and stunting condition. Authors should dedicate more space to the explanation of the null hypothesis, indeed.
Moreover in line 243-244 the authors reported that "the first time that the RT-qPCR machine was used to analyze oral Veillonella spp. from dental biofilm samples" it seems to be an incorrect declaration as other papers (an example is: Price RR, Viscount HB, Stanley MC, Leung KP. Targeted profiling of oral bacteria in human saliva and in vitro biofilms with quantitative real-time PCR. Biofouling. 2007;23(3-4):203-13. doi: 10.1080/08927010701251169. PMID: 17653931.) did the same analysis.
Another relevant point is to cite some of the newest cultural and genomic techniques for the analysis of oral microbiota dedicating a specific paragraph in discussion to them. Examples of papers to be cited can be:
1. Characterizing peri-implant and sub-gingival microbiota through culturomics. First isolation of some species in the oral cavity. A pilot study (doi:10.3390/pathogens9050365)
2. Dewhirst FE, Chen T, Izard J, Paster BJ, Tanner AC, Yu WH, Lakshmanan A, Wade WG. The human oral microbiome. J Bacteriol. 2010 Oct;192(19):5002-17. doi: 10.1128/JB.00542-10. Epub 2010 Jul 23. PMID: 20656903; PMCID: PMC2944498.
In the hope you'll be able to fix all the reported issues
Regards
Author Response
Response to Reviewer 2 Comments
Point 1: Dear authors, thanks for having submitted such an interesting manuscript.
My main concern is about the null hypothesis. Reading from the introduction the authors aimed at finding a sort of relationship between stunting and Oral Veillonella colonization. Considering that malnutrition is a well known causative factor for stunting and that Veillonella spp. can be found (even if not exclusively) in oral cavities of people with bad oral hygiene what seems to be lacking is the eventual relation between bacterial colonization and stunting condition. Authors should dedicate more space to the explanation of the null hypothesis, indeed.
Response 1: A previous study has stated that the number of oral Veillonella species significantly increased along with poor levels of oral hygiene. Pustelny et al. showed the presence of V. parvula could lead to a higher resistance of cocci bacteria, such as Streptococcus mutans, counter chemical antiseptics. In addition, deteriorating oral hygiene status in children may affect food selection and nutrient intake, leading to malnutrition. Many studies have been demonstrated the oral-gut microbiome axis can regulate the pathogenesis. In relation to Veillonella species, Zhan et al. reported that overabundance of Veillonella promote intestinal inflammation. It is evident, that Veillonella species might have role in pathogenesis yet the mechanism have not been fully clarified. (These has been stated in the 3rd paragraph)
Point 2: Moreover in line 243-244 the authors reported that "the first time that the RT-qPCR machine was used to analyze oral Veillonella spp. from dental biofilm samples" it seems to be an incorrect declaration as other papers. (an example is: Price RR, Viscount HB, Stanley MC, Leung KP. Targeted profiling of oral bacteria in human saliva and in vitro biofilms with quantitative real-time PCR. Biofouling. 2007;23(3-4):203-13. doi: 10.1080/08927010701251169. PMID: 17653931.) did the same analysis.
Response 2: We intend to report this study was the first time using species-specific primer on RT-qPCR machine due to increasing number of established oral Veillonella species such as V. atypica, V. denticariosi, V. dispar, V. infantium, V. parvula, V. rogosae, and V. tobetsuensis, and these information had been added.
Point 3: Another relevant point is to cite some of the newest cultural and genomic techniques for the analysis of oral microbiota dedicating a specific paragraph in discussion to them. Examples of papers to be cited can be: 1. Characterizing peri-implant and sub-gingival microbiota through culturomics. First isolation of some species in the oral cavity. A pilot study (doi:10.3390/pathogens9050365); 2. Dewhirst FE, Chen T, Izard J, Paster BJ, Tanner AC, Yu WH, Lakshmanan A, Wade WG. The human oral microbiome. J Bacteriol. 2010 Oct;192(19):5002-17. doi: 10.1128/JB.00542-10. Epub 2010 Jul 23. PMID: 20656903; PMCID: PMC2944498.
Response 3: This information has been cited at line 257-258

Reviewer 3 Report
The manuscript of Theodorea et al., is a well written manuscript and although the data are very few, they are interesting and obtained in humans.
- Lines 58-59: in my opinion weight should also be included in this definition.
- Figure 1: please use a defined color for the moderate stunted bar.
- Figure 2: caption is missing.
- Lines 94-95: please add information about how the sample was taken and how it was maintained
- Lines 193-194: something wrong in this sentence.
- Lines 243-244: in my opinion, considering that RT-qPCR isn't a cutting-edge technique, this sentence should be removed. A 16S sequencing would have given more consolidated data.
Author Response
Response to Reviewer 3 Comments
Point 1: The manuscript of Theodorea et al., is a well written manuscript and although the data are very few, they are interesting and obtained in humans. Lines 58-59: in my opinion weight should also be included in this definition.
Response 1: It has been added according to reviewer's suggestion
Point 2: Figure 1: please use a defined color for the moderate stunted bar.
Response 2: It has been revised
Point 3: Figure 2: caption is missing
Response 3: It has been revised
Point 4: Lines 94-95: please add information about how the sample was taken and how it was maintained
Response 4: It has been revised
Point 5: Lines 193-194: something wrong in this sentence
Response 5: It has been revised
Point 6: Lines 243-244: in my opinion, considering that RT-qPCR isn't a cutting-edge technique, this sentence should be removed. A 16S sequencing would have given more consolidated data.
Response 6: In the case of Veillonella species 16S sequencing by using 16SrDNA can not distinguish the genus Veillonella at the species level. It has been reported that it can be distinguished by using rpoB and dnaK.

Round 2
Reviewer 2 Report
Dear authors,
in may opinion the article is now acceptable in ita present form.
Regards